# The Effect of Telemonitoring (TM) on Improving Adherence with Continuous Positive Airway Pressure (CPAP) in Obstructive Sleep Apnoea (OSA): A Service Improvement Project (SIP)

**DOI:** 10.3390/healthcare10030465

**Published:** 2022-03-02

**Authors:** Abubacarr Gassama, Deyashini Mukherjee, Urwah Ahmed, Shirley Coelho, Mindi Daniels, Rahul Mukherjee

**Affiliations:** 1University Hospitals Birmingham NHS Foundation Trust, Birmingham B9 5SS, UK; a.gassama@nhs.net (A.G.); urwah.ahmed@nhs.net (U.A.); shirleykacoelho@gmail.com (S.C.); mindi.daniels@uhb.nhs.uk (M.D.); 2University Hospitals Coventry and Warwickshire, Coventry CV2 2DX, UK; deyashini93@hotmail.com; 3Institute of Clinical Sciences, University of Birmingham, Birmingham B15 2TT, UK

**Keywords:** OSA, CPAP, telemonitoring, adherence

## Abstract

The benefits of CPAP demonstrated in clinical trials are difficult to deliver in real life due to the lack of adherence. We analysed the effect of a Telemonitoring (TM)-related intervention on adherence as part of a Service Improvement Project (SIP) analysed as a retrospective cohort study. The ‘historical control’ (HC) cohort (followed up in conventional clinics) included all patients who commenced on CPAP between 1 February and 30 April 2019 (n = 142). The ‘telemonitoring’ (TM) cohort included all patients who commenced on CPAP between 1 May and 31 July 2019 (n = 166). Adherence was checked at 30 days (baseline) and 73 days for both cohorts. Wilcoxon—Rank test was used for statistical analysis (results reported as mean ± SEM). Both cohorts had similar adherence at the 30-day baseline, compared to a significantly lower adherence in the HC-cohort at 73 days (55.7 ± 3.0 vs. 51.8 ± 3.2% of days ≥ 4 h: *p* = 0.0072, average usage 255 ± 12.8 vs. 236 ± 13.7 min: *p* = 0.0003). There was a significantly higher adherence in the TM-cohort at 73 days (50.8 ± 2.5 vs. 56.1 ± 2.9% of days ≥ 4 h: *p* = 0.0075; average usage 234 ± 10.4 vs. 252 ± 12.1 min: *p* = 0.0456). Telemonitoring-feedback is effective at improving adherence with CPAP, suggesting its potential beneficial role in the community setting, particularly in the post-COVID reality of increased remote consultations.

## 1. Introduction

Obstructive Sleep apnoea (OSA) is a sleep-related respiratory disorder characterised by repetitive, partial, or complete collapse of the pharynx due to ineffective breathing effort, resulting in interruption of ventilation during sleep [1,2], causing sleep fragmentation and arterial hypoxemia. Untreated OSA may contribute to the pathophysiological mechanisms underlying the origin and development of cardiovascular diseases [3,4,5,6]. Continuous Positive Airway Pressure (CPAP) is the gold standard therapy for adults with moderate to severe OSA [7]. It works by delivering a continuous flow of room air through nasal or oronasal masks to pneumatically splint the upper airway and maintain patency, improving nearly all outcomes of OSA [8,9].

Despite the effectiveness of CPAP to treat OSA, adherence with therapy is a major issue for most patients [10]. Reasons for non-adherence include a dislike of CPAP, claustrophobia due to the mask, surgery (uvulopalatopharyngoplasty), noise, and discomfort of the apparatus [11,12]. It was reported that about 25–50% of patients abandon CPAP therapy within 4 weeks of treatment initiation [13,14]. Here, we define adherence as a composite of compliance and average usage. A patient is deemed compliant if the average usage is ≥4 h and at least 70% of the nights have ≥4 h usage [10], the level required to gain meaningful benefit from treatment with respect to reduction in symptoms of daytime sleepiness, and improved health-related quality of life, mood, and attendance at work. It is important to use this composite figure as average usage alone may not accurately reflect the adherence to therapy (i.e., the number of nights that the patient actually attempts to use CPAP), particularly in patients with unhealthy lifestyles [15,16].

Four years ago, we embarked on a service improvement project (SIP) to improve adherence with CPAP therapy in OSA patients. The first cycle of this SIP investigated the average daily usage of patients completing annual follow-up after initiation of CPAP for OSA [17]. We used the apnoea-hypopnoea index (AHI) to categorise severity: normal was <5, mild was 5–14, moderate was 15–29, and severe was 30 and above. We found that we had limited capacity for face-to-face follow-up as our centre is a high-volume centre (more than 700 CPAP issues per year). This meant that we were only able to follow up patients with severe OSA or patients who were heavy goods’ vehicle drivers. Moderate and mild sufferers were missing out on follow-up. The first cycle recommended implementing Telemonitoring (TM) into our follow-up arrangements to identify those struggling with treatment and offer resolutions to problems encountered during the early phase of treatment initiation [17], which is critical as patients’ CPAP usage during the first two weeks may predict long-term concordance to treatment [8]. We decided to continue our SIP by performing a second retrospective audit to assess the effect of a TM-related personalised intervention by a physiologist on CPAP adherence.

## 2. Methods

TM was offered to all new patients initiated on CPAP in the sleep service at the Birmingham Heartlands Hospital (BHH) between 1 February 2019 and 31 July 2019 (477 patients), out of which 308 patients (64.5%) met both necessary criteria of having the infrastructure (domestic broadband internet) and consenting to being monitored by cloud remote monitoring (TM). We collected CPAP compliance data on all 308 patients eligible by the above criteria. The patients who started on CPAP in the first three months of the study acted as the ‘historical controls’ (142 patients) and were initiated on the standard departmental protocol (passive remote monitoring without intervention). Those set up on CPAP during the latter three months became the ‘intervention arm’ (166 patients) and underwent telemonitoring with physiologist intervention at 4–6 weeks (average 30 days) after initiation of CPAP (Figure 1). The intervention included a telephone consultation, or a letter based on the TM data to help resolve the following issues: (1) adjusting CPAP pressures, (2) mask re-fitting including offering alternative masks, and (3) offering a humidifier. This was the TM-related personalised intervention by a physiologist as part of a service improvement drive, which applied to all consenting service users. We re-assessed the CPAP usage after a “cooling down” period of 4–6 weeks after the intervention. We used the 73rd day (the middle of the 11th week after CPAP initiation) as the end point to assess the effectiveness of the intervention.

Primary outcomes were CPAP compliance (use of CPAP % for ≥4 h on ≥70% of total nights) and average usage (minutes). Patient demographics and baseline characteristics were obtained retrospectively from patients’ clinical letters at the time of OSA diagnosis, including AHI, Epworth Sleepiness Scale (ESS), and Body–Mass Index (BMI). In addition, CPAP machine return records were quantified for both groups to assess acceptance of CPAP therapy (Table 1).

Baseline measurements (first data collection point) of compliance, adherence, and average usage were recorded at 30 days after CPAP initiation for both control and intervention groups. The intervention group was reviewed via telemonitoring 4–6 weeks post-CPAP therapy initiation by a respiratory physiologist. This involved a phone call or letter, change in pressure, resolving mask issues (e.g., advice on fitting, leakage reduction, maintenance matters, or attendance at mask clinic drop-in sessions), or invitation to the respiratory department for more complex matters. The average time for the intervention for all the patients was 43 days (Figure 2). Therefore, to make the second data collection point in the control arm comparable to the intervention group, the average intervention period of 43 days was added to the initial 30 day, totaling 73 days. At this point, 30-day compliance and average usage were recorded for the control (Figure 2).

Statistical analysis was performed using GraphPad Prism 5 software or SPSS for windows version 20 (IBM SPSS). Standard descriptive statistical analysis was completed to look at the general characteristics of the study participants and Kolmogorov—Smirnov tests were used to assess the normality and variance of the variables, respectively. Wilcoxon—Rank test was used to compare the outcome variables pre- and post-intervention for both the control and TM group. Mann—Whitney U test was used to compare the control versus intervention group, which showed no statistically significant difference between the two groups. Variability values were presented as 95% confidence interval (CI) or median, 75th, and 25th percentile. Statistical significance was taken as *p* < 0.05.

The power calculation indicated that in order to capture a difference of 0.95 in CPAP compliance at a significant level of 5% and with a power of 95%, we required a total of 57 participants (G*Power 3.1.9.2) (Heinrich Heine Universitat, Dusseldorf, Germany). Remote monitoring data was obtained from a total of 308 patients. Therefore, our sample size was powered to predict statistical variability in our data.

## 3. Results

A total of 142 participants were included in the historic control arm (HCA), while 166 participants were included in the intervention arm (IA). The median age for the control and intervention groups were 49 (24–74) and 52 (21–83) years, respectively (Table 2). The median of ESS (12 versus 13), BMI (35.5 versus 35.85), and AHI (31.5 versus 31.7) were not significantly different between the groups. Thus, the groups were matched, and any outcome variation is not due to baseline predisposition (Table 2).

### 3.1. Historic Control Arm

In the HCA, without TM and physiologist intervention, both compliance (≥4 h usage per day; %) with CPAP therapy and its average usage (minutes) decreased from the baseline (measured at 30 days post-CPAP initiation) to 73 days post-CPAP initiation.

At the baseline, mean compliance was 55% (95% CI: 49.7–61.60) compared to 51.82% for the controls (95% CI: 45.44–58.21; *p* value < 0.0072). The box plot demonstrates that although the median and 75th percentile were both at 60% of nights with ≥4 h for both the baseline and control, a significant number of patients moved into the lower 25 percentile at the second measurement point than at the baseline, indicating diminution of therapy compliance.

Similarly, the mean average usage decreased from 255 min (95% CI: 229.7–280.2) at the baseline to 236.1 min at 73 days (95% CI: 209.1–263.2; *p* value < 0.0003). In addition, the median, 75th percentile, and 25th percentile of the average usage (minutes) all significantly decreased from the baseline (261.5, 383, and 118) compared to the control (254, 371, and 58.5), respectively (Figure 3). Again, a significant number of users moved into the lower 25th percentile without TM.

### 3.2. Intervention Arm

In the IA, compliance with CPAP therapy improved from the baseline to after TM (Figure 4). The mean compliance increased from 50.84% (95% CI: 45.85–55.83) to 56.10% (95% CI: 50.40–61.79; *p* value < 0.0075). The median and 75th percentile of the compliance both improved after the intervention (63% and 93%) compared to the baseline (53% and 80%), respectively (Figure 4). Although the 25th percentile decreased slightly in the intervention compared to the baseline (17% vs. 18.5%), a significant number of people moved up into the upper 25th percentile (Figure 4). Chen et al.’s recent meta-analysis of 11 studies involving a total of 1,332 CPAP users, concluded a significant improvement in compliance in the TM group (mean difference 0.68 h, 95% confidence interval 0.48–0.89 h) vs. the control group [18]. There was low to moderate heterogeneity within the studies included in the meta-analysis as indicated by the *I*^2^ value (49%) [18].

Similarly, the average usage in minutes also increased after TM compared to the baseline. The mean average usage increased from 213.5 min (95% CI: 213.5–254.5) at the baseline to 252.3 min (95% CI: 228.4–276.2; *p* value < 0.0456) post-TM. Many patients moved from the lower quartiles into the upper 25th percentile post-TM in the intervention group (Figure 4). The median and the 75th percentile of the average usage increased in the post-TM (266.5 and 380.5) compared to the baseline (234 and 325.5), respectively (Figure 4). Our findings are consistent with a 2016 study by Clavaud and Cooper [10], which revealed a significantly longer daily CPAP usage in remote monitoring (12 weeks) with physiologist intervention at 1, 4, and 8 weeks versus the controls (504 ± 0.0 vs. 460 ± 0.4 min; *p* < 0.01; mean ± SEM). In addition, a randomised clinical trial by Stepnowsky et al. (2013) [19] used an interactive internet-based platform that offered sleep service providers access to remotely monitor treatment and to troubleshoot CPAP related problems. The study found that TM increased average CPAP usage by 1 h/night compared to the control at 2-month and 4-month post-CPAP initiation [19]. The same group previously showed improved average usage (hours/night) in the TM group (4.1 ± 1.8) compared to usual care (2.8  ±  2.2) in a pilot study of 45 CPAP users, but the difference was not significant [20]. We noted a high statistical error level in both groups even though the average usage was increased by at least 1.3 h/night.

There was an increased number of CPAP users that accepted therapy in the TM group compared to the control group with 68% and 61% continuing the therapy, respectively. A total of 34 patients (15%) in the historic control group returned their CPAP machines, compared to 25 (10%) patients in the intervention group. These differences did not reach statistical significance.

## 4. Discussion

Our service improvement project firstly estimates the feasibility of TM for CPAP therapy in an urban population (about two-thirds of CPAP users met both criteria of having the domestic infrastructure for TM and consenting to it). With passage of time, this proportion is expected to increase as broadband internet becomes more affordable. Among those able to participate in TM, our results clearly demonstrate improved adherence to CPAP with the TM-related personalised intervention by a physiologist. The term adherence ‘emphasises the need for agreement and that the patient is free to decide whether or not to adhere to the doctor’s recommendation’ [21]. It does not take into account whether the patient is able to adhere to therapy. This is particularly important with a physical therapy such as CPAP, where a patient may suffer initial problems (outlined in the introduction). TM enables earlier detection of factors that may perpetuate non-adherence, hence allowing earlier resolution of these issues (e.g., mask adjustments, machine settings, etc.). This should in turn improve concordance, a more modern term that focuses more on patient support and the doctor-patient relationship [21]. Going forward, we hope to see the impact of TM on concordance with CPAP in OSA in a quantifiable manner.

Although the old departmental protocol routinely offered walk-in access to all CPAP patients for mask-related issues, patients tended to wait for their scheduled follow-up appointment (6 months post-CPAP initiation) to seek advice on any issues. By this time, several patients may have already given up [13,14]. TM can aid prioritisation of follow-up appointments to improve adherence early on when most gains are to be made. Furthermore, this can also filter out patients who are not using the machine and/or will gain no benefit from treatment. This in turn allows us to retrieve unused CPAP machines and recycle these machines amongst those who need it and will use it, thereby ensuring efficient resource allocation. Additionally, this will relieve patients from a burdensome therapy that they are not gaining benefit from.

Increasing follow-up capacity is a pertinent issue in the context of the COVID-19 pandemic, when face-to-face appointments have been scarce, and may continue to be so going forward. Reduced footfall on healthcare facilities has been shown to reduce the spread of respiratory viruses. Therefore, TM is crucial from an ongoing infection control perspective.

TM has the potential to play a role in many other aspects of the OSA management pathway outside follow-up [22]. This would start from the diagnostic stage with remote respiratory polygraphy and polysomnography [22], including patient education, which has been shown to improve CPAP adherence [15]. We aim to implement TM earlier in the OSA management pathway as we go forward, looking at its impact on patient outcomes and satisfaction.

Incorporating TM in the follow-up of patients on CPAP therapy may widen access to CPAP in patients who under normal circumstances may be less able to attend face-to-face follow-up consultations [23]. However, as with interventions relying on technology, the health inequality gap could be widened for those who cannot access technology. This is a new area that needs to be further explored in future studies as the use of TM expands, particularly in the post-COVID world.

There is mixed evidence regarding the impact of CPAP on comorbidities. Increased CPAP adherence may see improvement in glycaemic control [24,25] and cardiovascular disease, particularly for severe OSA patients [6,12,26]. Our first cycle showed a significant association between cardiac comorbidities (arrhythmia and congestive heart failure) and CPAP adherence [17]. In adjusted analysis, a multicenter study conducted in Spain that compared CPAP with usual care in 725 patients with OSA who did not have prior cardiovascular disease, showed better outcomes among patients who were adherent to CPAP therapy (≥4 h per night) than among patients who did not receive CPAP or who used CPAP less than 4 h per night [12]. Further studies are needed to see whether improved CPAP adherence with a TM-related personalised intervention by a physiologist also improves associated comorbidities and cardiovascular outcomes. The most important limitation has been the access to remote monitoring only for 3 months post-CPAP initiation at the time of this project, which made a longer period of remote monitoring unfeasible. Consequently, the study was designed to capture the data within 90 days of CPAP initiation. Therefore, whilst we have established that TM and physiologist intervention increases compliance and average usage of CPAP in the first 10 weeks of treatment initiation, our study cannot comment on whether this would be maintained over a longer period of time or if the gains in compliance and average usage would be lost over time. In the month of May 2020, our department signed an extended license to access remote monitoring for at least a year. This will give us an opportunity in the future to engage in longer-term monitoring combined with physiologist intervention, which can be directly compared with the 6 months CPAP follow-up.

## 5. Conclusions

Our SIP shows that implementation of TM with active intervention by a physiologist in the first 4–6 weeks leads to a statistically significantly better compliance and increased average usage, thus better adherence to CPAP in OSA patients across all AHI groups. There is further scope of implementation of the TM-related personalised intervention by a physiologist earlier in the OSA management pathway to help deliver meaningful benefit from treatment, specifically with respect to reduction in symptoms of daytime sleepiness, improved health-related quality of life, mood, and attendance at work. TM also has the potential to facilitate further studies of the effect of CPAP adherence on OSA-associated comorbidities. TM will be an essential tool for monitoring OSA patients in a post-COVID world with limitations on face-to-face interaction to meet infection control measures.

## Figures and Tables

**Figure 1 healthcare-10-00465-f001:**
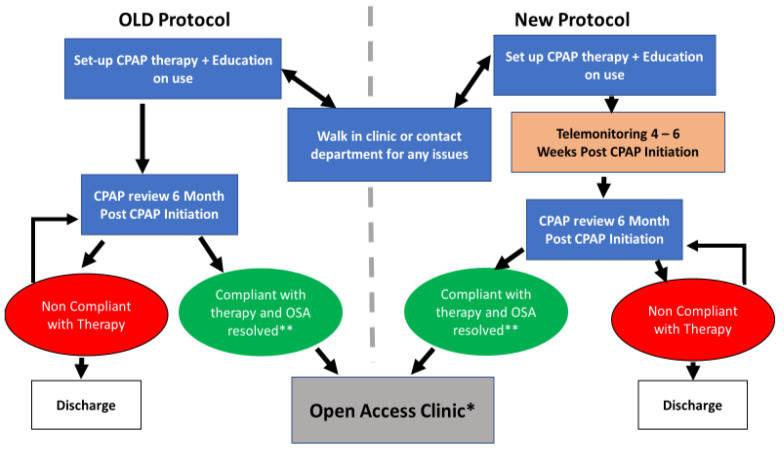
Illustration of the CPAP therapy management pathway; The old and new protocols were applied on the historical control and intervention arms in this service improvement project. OSA = obstructive sleep apnoea; * This patient does not have a routine CPAP follow-up appointment but can still access the CPAP walk-in clinic or get in touch with the physiology department for any issues with the therapy; ** Compliant with unresolved OSA might require further investigation or medical review.

**Figure 2 healthcare-10-00465-f002:**
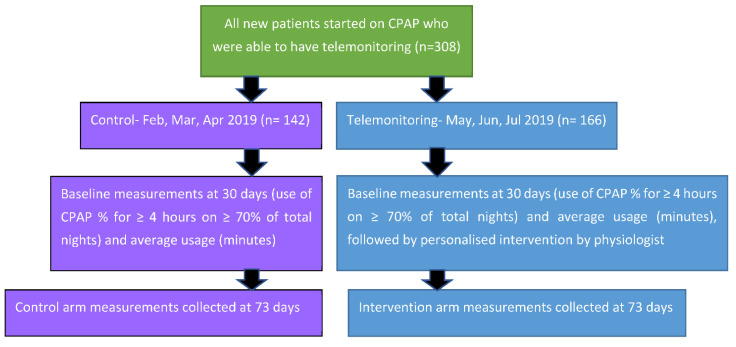
Schematic representation of the design of the Service Improvement Project and timeline of measurements.

**Figure 3 healthcare-10-00465-f003:**
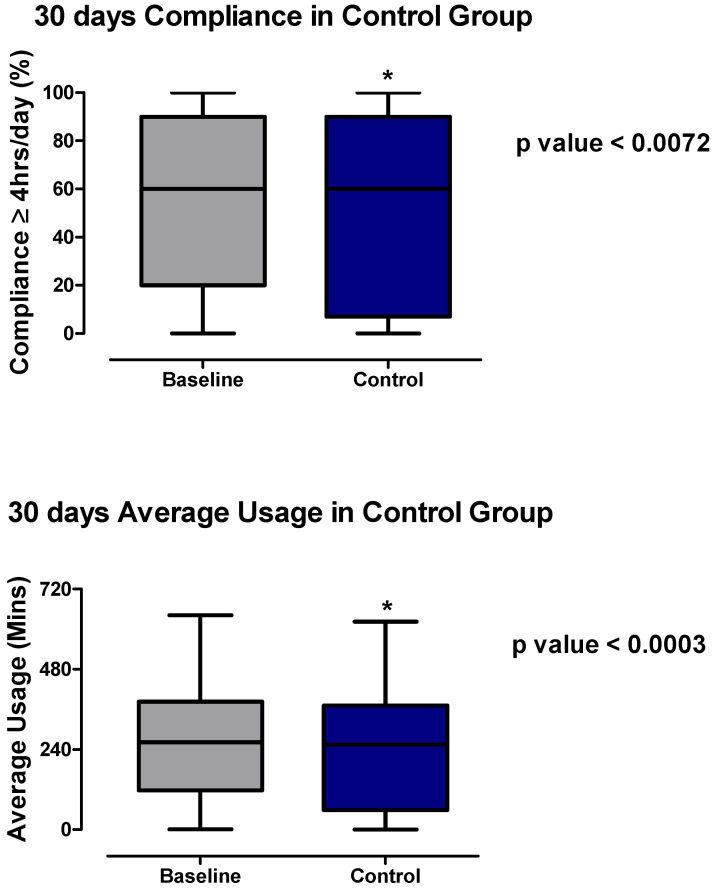
Average usage per day (minutes) and compliance (≥4 h usage day %) in controls at two time points: 30 days before (baseline) and 30 days after the telemonitoring phase (control). Wilcoxon—Rank test was used to compare the group. * *p* value ≤ 0.05 was considered to be significant.

**Figure 4 healthcare-10-00465-f004:**
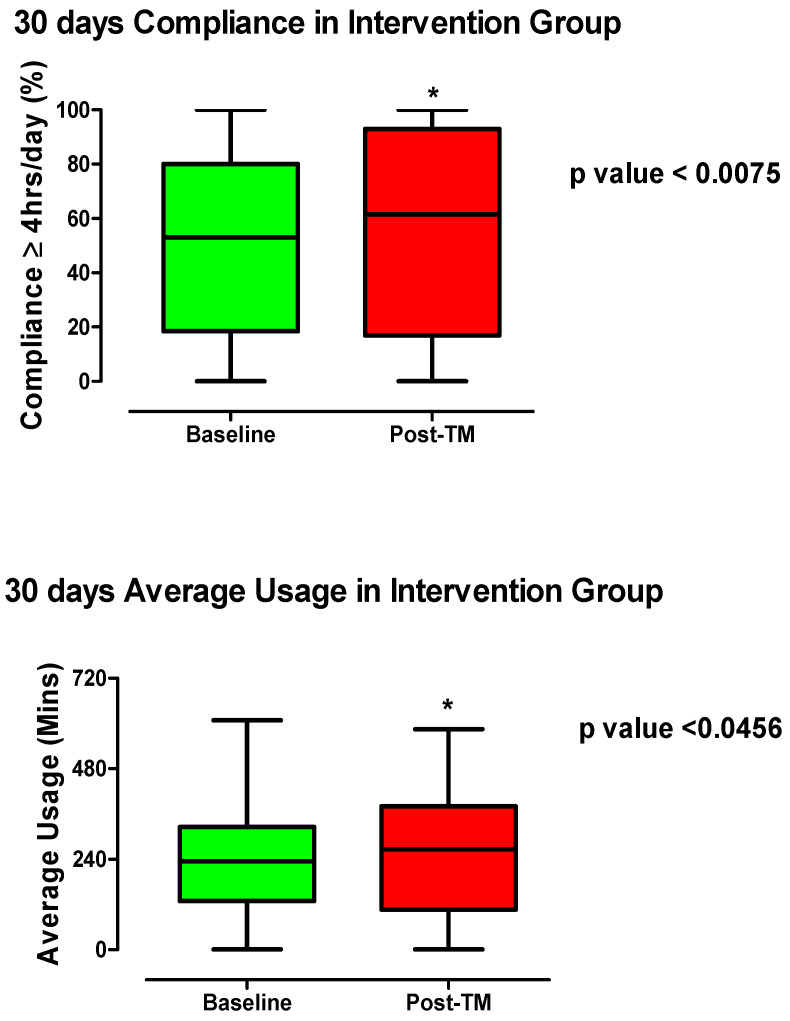
Average usage (in minutes) and compliance (≥4 h usage day %) pre- (30 days post-CPAP initiation) and 30 days post-telemonitoring intervention. Wilcoxon—Rank test was used to compare the group. * *p* value ≤ 0.05 was considered to be significant.

**Table 1 healthcare-10-00465-t001:** Breakdown of CPAP data information and machine returns.

	Controls (% of Total)	Intervention (% of Total)
Patients able to have Remote monitoring	142 (61)	166 (68)
Returned Total	34 (15)	25 (10)
Patients unable to have Remote monitoring	56 (24)	54 (22)
Total CPAP Set-ups	232	245

**Table 2 healthcare-10-00465-t002:** Patient demographics and baseline characteristics; The control and intervention groups were matched. No significant difference was observed in age, gender, BMI (body mass index), ESS (Epworth sleepiness score), and AHI-Dx (apnoea-hypopnoea index at diagnosis) between the two categories at the time of diagnosis. C = control; TM = Telemonitoring.

	Number of Values	Minimum	25th Percentile	Median	75th Percentile	Maximum
Gender-C (M/F)	142 (92/50)					
Age-C	142	24	40	49	59	74
ESS-C	137	0	8	12	16	24
BMI-C	128	23.6	31.43	35.5	40.78	59.2
AHI-Dx-C	142	8.7	18.68	31.5	52.83	161.8
Gender-TM (M/F)	166 (95/71)					
Age-TM	166	21	40	52	60.25	83
ESS-TM	146	0	8	13	17	24
BMI-TM	156	19	29.43	35.85	43.83	82.8
AHI-Dx-TM	161	3.1	17.95	31.7	52.4	146

## Data Availability

The data supporting the reported results are held by the department of Respiratory Medicine & Physiology at the Birmingham Heartlands Hospital and all authors have full access to the data. Part of the data was a spoken presentation as an abstract titled “The effect of telemonitoring on improving concordance with continuous positive airway pressure (CPAP) in obstructive sleep apnoea (OSA)” at the 2020 Winter Meeting of the British Thoracic Society held in February 2021 (http://dx.doi.org/10.1136/thorax-2020-BTSabstracts.136; accessed on 27 February 2022).

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
