# Peer review of "The Effect of Telemonitoring (TM) on Improving Adherence with Continuous Positive Airway Pressure (CPAP) in Obstructive Sleep Apnoea (OSA): A Service Improvement Project (SIP)"

_healthcare, 2022, doi:10.3390/healthcare10030465_

Round 1
Reviewer 1 Report
46- I would reconsider stating that 4h/night for 70% of the night derives meaningful benefit from treatment. The reference for this statement is another patient that asks the same clinical question as your paper; it is not a reference that describes what meaningful benefit is. I suggest finding a paper that supports this assertion.
71- major weakness for this paper. 477 patients were included in the study but only 308 had data? Potentially, 35% of the cohort could simply not be using PAP.
For this paper to have valid results, statistics should be run on the entire cohort. Alternatively, if the authors can explain why 35% of the original cohort is not analyzed (find those that are not compliant vs. those that didn't have cloud monitoring), then it may be seen as valid.
Finally, while your study finds that remote monitoring statistically improves compliance, I think it would be useful to discuss if might be clinically significant. Your discussion references that PAP improves glycemic control and CV outcomes. I recommend specifying the required compliance time to reach those improved outcomes and compare whether the improvement you see in your study would reach clinical significance.
Author Response
46- we have modified the text (lines 47-48 in the revised manuscript) to reflect that the "≥ 4 hours and 70% of night with ≥4 hours" is the level of usage required to gain meaningful benefit from treatment specifically with respect to reduction in symptoms of daytime sleepiness, improved health-related quality of life, mood, and attendance at work. This is to reflect the fact that it is unclear whether this level of usage has benefits with respect to other outcomes, especially cardiovascular. Furthermore, we have also switched reference 12 to: Barbé F, Durán-Cantolla J, Sánchezde-la-Torre M, et al. Effect of continuous positive airway pressure on the incidence of hypertension and cardiovascular events in nonsleepy patients with obstructive sleep apnea: a randomized controlled trial. JAMA 2012;307:2161-8. This is a multicenter study conducted in Spain that compared CPAP with usual care in 725 patients with OSA who
did not have prior cardiovascular disease. In adjusted analysis, this study reported better outcomes among patients who were adherent to CPAP therapy (≥4 hours per night) than among patients who did not receive CPAP or who used CPAP less than 4 hours per night.
71- we thank reviewer 1 especially for this comment. Our service improvement project explored (a) the feasibility of telemonitoring and (b) the effect of a telemonitoring-related personalised intervention by a physiologist on adherence/compliance with CPAP. We have reflected this through a major rewriting of the Methods section (lines 65-84), Discussion section (lines 193-198) and Table 1 in the revised manuscript, to make the information absolutely clear, in response to this comment.
"I recommend specifying the required compliance time to reach those improved outcomes and compare whether the improvement you see in your study would reach clinical significance."
We have added lines 266-269 in the revised manuscript to reflect the fact that our target usage is known to have benefits with respect to reduction in symptoms of daytime sleepiness, improved health-related quality of life, mood, and attendance at work but it is unclear whether that improves other outcomes, especially cardiovascular.

Reviewer 2 Report
The effectiveness of CPAP to treat OSAS, Adherence with therapy is important for all patients. This study really demonstrate improved adherence to CPAP with TM.
It is true that is a new area that needs to be further explored in future studies as the use of TM expands, particularly in the post-COVID world.
This paper would enable us to see whether improved CPAP adherence with TM also improved associated morbidities.
Author Response
We thank reviewer 2 for taking the time to read our manuscript and grateful for the supportive comments. We have modified lines 269-270 in the Conclusion section of the manuscript to reflect that telemonitoring has the potential to facilitate further studies of the effect of CPAP adherence on OSA-associated comorbidities.

Reviewer 3 Report
I am thankful for giving me the chance to review the manuscript entitled“The Effect of Telemonitoring (TM) on Improving Adherence 2 with Continuous Positive Airway Pressure (CPAP) in Obstruc-3 tive Sleep Apnoea (OSA): a Service Improvement Project (SIP)”. The clinical significance is existed, the method is reasonable, and the reporting is good. I have only few concerns.
- The details of telemonitoring should be addressed.
- Why do you choose 73 days as the end point?
- What do you mean “average #days before intervention = 43” in Figure 2?
- “Gender-RM” should be changed to “Gender-TM” in Figure 3. You should point out what test was used to show no difference between the 2 groups. In addition, Figure 3 should be presented as Table 2.
- I suggest to delete “Shows” in Figure 4 and Figure 5.
Author Response
We thank reviewer 3 for taking the time to read our manuscript and for their comments. Please find our point-by-point response below:
- The details of telemonitoring should be addressed:
We have entered the specific description of the telemonitoring-related personalised intervention by a physiologist. We have modified lines 81-86 in the Methods section to clearly define what the intervention entailed. - Why do you choose 73 days as the end point?
As described in the Methods section, the TM-related personalised intervention by a physiologist was conducted 4-6 weeks (average 30 days) after initiation of CPAP. We wanted to assess the CPAP usage after a "cooling down" period of 4-6 weeks. This is why we wanted to check the usage in the 11th week. The 73rd day is in the middle of the 11th week, hence we used this as the end point to assess the effectiveness of the TM-related personalised intervention by a physiologist. This is explained in lines 81-89 of the revised manuscript. - What do you mean “average #days before intervention = 43” in Figure 2?
We have modified figure 2 to clarify our methodology as outlined in point 2 above (please see lines 81-89 of the revised manuscript). - “Gender-RM” should be changed to “Gender-TM” in Figure 3. You should point out what test was used to show no difference between the 2 groups. In addition, Figure 3 should be presented as Table 2.
The table has been altered and says "Gender-TM". Figure 3 has now been rendered as Table 2 in the revised manuscript. Please note that we have been unable to delete the old figure 3 (in orange) but we have inserted Table 2 (in blue). As mentioned in lines 119-120 of the revised manuscript, Mann-Whitney U Test was used to compare control versus intervention groups, which showed no statistically significant difference between the two groups. - I suggest to delete “Shows” in Figure 4 and Figure 5.
We have deleted "Shows" from both of these figure legends. Please note that as the previous Figure 3 is now Table 2 in the revised manuscript, the old Figures 4 and 5 have been renamed Figure 3 and Figure 4 respectively.
